# An Energy-Efficient and Obstacle-Avoiding Routing Protocol for Underwater Acoustic Sensor Networks

**DOI:** 10.3390/s18124168

**Published:** 2018-11-27

**Authors:** Zhigang Jin, Mengge Ding, Shuo Li

**Affiliations:** 1School of Electrical and Information Engineering, Tianjin University, Tianjin 300072, China; mengge_ding@163.com; 2School of Engineering, RMIT University, Melbourne, VIC 3000, Australia

**Keywords:** underwater acoustic sensor networks, routing protocol, energy-efficient, obstacle-avoiding, fuzzy logic-based forwarding relay selection

## Abstract

Underwater Acoustic Sensor Networks (UASNs) have become one of the promising technologies for exploring underwater natural resources and collecting scientific data from the aquatic environment. As obstacles hinder the communications among sensor nodes in UASNs, designing an effective bypass routing protocol to avoid obstacles is an urgent need. Moreover, the sensor nodes are typically powered by batteries, which are difficult to replace, restricting the network lifetime of UASNs. In this paper, an Energy-efficient and Obstacle-Avoiding Routing protocol (EOAR) is proposed not only to address the issue of marine animals acting as obstacles that interfere with communications, but also to balance the network energy according to the residual energy. In the EOAR protocol, when the current node perceives the existence of marine animals, the interference area of the animal-nodes is first calculated using the underwater acoustic channel model, and then the candidate forwarding relay set of the current node is obtained according to the constraint conditions. The optimal candidate forwarding relay is determined by a fuzzy logic-based forwarding relay selection scheme based on considering the three parameters of the candidate forwarding relay, which includes the propagation delay, the included angle between two neighbor nodes, and the residual energy. Furthermore, in order to solve the problem of energy waste caused by packet collision, we use a priority-based forwarding method to schedule the packet transmission from the candidate forwarding relay to the destination node. The proposed EOAR protocol is simulated on the Aqua-sim platform and the simulation results show that proposed protocol can increase the packet delivery ratio by 28.4% and 11.8% and can reduce the energy consumption by 53.4% and 32.7% and, respectively, comparing with the hop-by-hop vector-based forwarding routing protocol (HHVBF) and void handling using geo-opportunistic routing protocol (VHGOR).

## 1. Introduction

Underwater Acoustic Sensor Networks (UASNs) have proven their potential in a variety of aquatic applications, such as marine environmental monitoring, resource exploration, marine animal studies and assisted navigation [1,2]. 

However, due to the complex underwater environment, realizing effective communication among sensor nodes in the UASN is much more difficult than those in traditional terrestrial wireless sensor networks (TWSNs) [3]. The reason is that radio waves employed in the terrestrial sensor networks are not feasible in underwater communications because of the rapid attenuation of the electromagnetic radio waves and the sharply decrease of the communication range [4]. Currently, the only feasible method for long range communication in underwater environments is acoustic signals, which can cover long-time and large-scale underwater communications. However, the speed of sound is approximately 1500 m/s underwater, five-orders slower than the speed of radio waves, which causes a 0.67 s/km propagation delay in UASNs [5]. 

In order to overcome the above limitations and make aquatic applications viable, efficient communication protocols, especially the routing protocols, are essential for successful communications. Geographic routing can work together with opportunistic routing (geo-opportunistic routing) [6] to improve packet delivery ratio. Using the geo-opportunistic routing paradigm, every sensor node in the UASN is aware of its own location. A data packet is broadcast by its source node to the candidate forwarding relay set composed of neighbors of this node first. Then the nodes in candidate forwarding relay set are ordered according to some metrics which define their priorities. In such a way, the candidate forwarding relays broadcast the received packet one by one with a small time-gap in order of priorities. If a lower priority forward relay overhears the same packet forwarded by a higher priority forwarding relay before it starts to broadcast this packet, this relay will discard the packet and will not broadcast it anymore. This procedure is repeated until the packet finally reach the destination node. Such kind of multi-node participation in forwarding can guarantee the packet delivery ratio. However, if multiple nodes participate in the data forwarding procedure, the network energy consumption will increase sharply because of a large amount of information transmission and reception. Therefore, the energy consumption issue has a major impact on designing the routing protocol.

Besides, the lifetime of UASNs is restricted by the limited energy due to the difficulty to replace the batteries of the underwater sensor nodes. How to balance the energy of the sensor nodes in order to extend the network lifetime has become a hot issue in UASN research [7]. In [8], a location-free reliable and energy efficient pressure-based routing protocol (RE-PBR) is proposed, which defines the route cost based on link quality and residual energy of the sensor nodes. The forwarding relay selection of RE-PBR takes the residual energy of sensor nodes into account, so that the lifetime of the network can be extended. However, sensor nodes are deployed in UASNs sparsely, and most of the nodes are moving because of water currents or any other underwater action, which causes the problem of void area. The routing protocols like RE-PBR cannot solve the problem of avoiding routing void, leading to the need for an adaptive routing protocol that is able to handle the void area problem.

In recent years, to handle the void area problem, many routing protocols [9,10,11] have been proposed. In [9], convex void handling and concave void handling based on VHGOR are proposed to bypass the void area. This method can adaptively determine the reconstruction way of routing path according to whether the void encountered by the node is convex or concave and effectively address the void crisis. However, the coexistence of acoustical multi-systems in UASNs brings new challenges, especially obstacle avoidance, to underwater routing protocols. In particular, some marine animals as part of a natural acoustic system also use acoustic signals to communicate, causing interfere to communications among sensor nodes around the marine animals [10,11]. The routing protocols which are able to avoid the disruption to the routes in UASNs caused by marine animals (considered as obstacles) are called obstacle-avoiding routing protocols, which are special void handling routing protocols. Currently, most of the existing underwater acoustic routing protocols, e.g., VHGOR, focus on prolonging the network lifetime and handling the traditional void area problem, but only a few routing protocols committed to solved the obstacle-avoiding problem. Therefore, propose an effective routing protocol that not only avoids the obstacle but also prolongs the network lifetime, for UASNs, is very crucial. 

To solve the issues above in UASNs, in this paper, we design an energy-efficient and obstacle-avoiding routing protocol (EOAR), which mainly contains three steps. Firstly, when a current node perceives the presence of marine animals during the underwater communications, the interference area of the animal-nodes and the candidate forwarding relay set are first calculated using the underwater acoustic channel model [12]. Then, the fuzzy logic-based relay selection scheme, taking the propagation delay, the included angle between two neighbor nodes, and the residual energy into consideration, is applied to calculate the forwarding probability of the candidate forwarding relays. Last, a priority-based forwarding method, which can avoid the packet collisions, is applied to schedule the packets transmission towards the destination node. The following are the key contributions of this paper:
(1)In a network where marine animals and sensor nodes coexist, the underwater acoustic channel model is used to determine the interference area of the animal-nodes and then form the candidate forwarding relay set, which can reduce the possibility of those sensor nodes that are interfered by marine animals becoming the candidate forwarding relay set.(2)A new routing protocol call EOAR is proposed in this paper. In EOAR, the forwarding order of the nodes belonging to the candidate forwarding relay set is sorted using a fuzzy logic-based forwarding relay selection scheme. By considering more metrics, including propagation delay, the included angle between two neighbor nodes and the residual energy, the proposed EOAR protocol improves packet delivery ratio and extends network lifetime compared to routing protocols previously applied to fuzzy logic [13].

The simulation results show that our protocol can avoid animals’ interference more effectively, increases the packet delivery ratio by 28.4–11.8%, and reduces energy consumption by 53.4–32.7% compared with the classic hop-by-hop vector-based forwarding routing protocol (HHVBF) and void handling using geo-opportunistic routing protocol (VHGOR).

The rest of this paper is organized as follows: in Section 2, related work on underwater routing protocols is discussed briefly. In Section 3, the proposed network model is introduced. In Section 4, the EOAR protocol is described in detail. The simulation results are shown and discussed in Section 5. Finally, this paper is concluded in Section 6.

## 2. Related Work

In UASNs, the oceanographic data collection is often completed by communication among nodes. An efficient communication is inseparable from the support of routing protocols. In recent years, the design of acoustic network routing protocol has gradually become a hot topic in underwater acoustic network research [14,15], and has been committed to solve the unique characteristic of UASNs such as high energy consumption, high-latency and low packet delivery ratio. In this section, we provide a review on research works on this topic.

The Vector-Based Forwarding (VBF) protocol [16] is a classical underwater acoustic routing protocol for three-dimensional underwater sensor networks. The protocol establishes a cylindrical routing pipeline between the source node and the destination node. Only the nodes in the pipeline can participate in packet forwarding. This protocol can select the optimal nodes to forward data in a network with many sensor nodes, thereby saving energy consumption. In [17], hop-by-hop vector-based forwarding routing protocol (HHVBF) is proposed. In HHVBF, each forwarding relay that performs forwarding establishes a virtual routing pipe between the itself and the destination node. Thus, HHVBF divides the long pipe in the VBF into several short pipes to forward data. Besides, HHVBF and VBF protocols both set a waiting time for each forwarding relay to avoid the packet collision, thereby further saving energy consumption.

In [18], the authors proposed a depth-based routing (DBR) protocol for underwater networks. DBR uses the depth sensors to perceive depth information for simple greedy forwarding. The forwarding relay with lower depth is selected to forward the packet, which increases the packet delivery ratio but also consumes much more energy. Many proposed routing protocols, like DBR, have high energy consumption. However, the power of the sensor nodes is limited, so the energy problem has gradually become a key challenge of the underwater acoustic routing protocol.

In [13], a power-efficient routing protocol with the aid of the fuzzy logic system and decision tree, which determines the appropriate forwarding relay to forward the packets to the destination node, is proposed. This routing protocol uses a tree trimming mechanism to prevent the growth of the packet forwarding tree, thereby effectively reducing the energy consumption of the sensor nodes.

Besides, recharging the battery for a sensor node deployed under water for communication is a very laborious task [19], so the limited energy restricts the network lifetime of UASNs, which makes it important to find routing protocols that extend the network lifetime. Energy Balanced and Lifetime Extended Routing (EBLE) protocol is proposed in [20]. This protocol, which selects the forwarding relays with high residual energy and relatively low-cost paths based on the cost function and residual energy level information, not only balances the traffic load with the remaining energy but also optimizes the data transmission by selecting the routing path with low cost. In [21], Q-learning-based delay-aware routing (QDAR) algorithm utilizes a Q-learning algorithm to extend the network lifetime for UWSNs. QDAR algorithm defines an action-utility function in which propagation delay ratio and residual energy are both considered for acoustic routing decisions, and thus can prolong the network lifetime by distributing the residual energy.

Moreover, sensor nodes are usually sparsely deployed in UASNs, and movement of the sensor nodes caused by water currents is inevitable, which cause the void problem. In order to reduce the probability of encountering void holes in the sparse networks, the protocols proposed in [22,23] can reduce the void hole problem during forwarding path by selecting forwarding relays based on the information on the number of two-hop potential neighbors. In [24], the authors proposed weighting depth and forwarding area division DBR routing protocol (WDFAD-DBR). WDFAD-DBR not only reduces the probability of encountering void holes in the local sparse area but also saves the energy consumption. 

However, these routing protocols basically do not consider the problem of obstacle avoidance. In particular, marine animals which use the acoustic signals to communicate interfere with communication among sensor nodes. In this paper, we propose an energy-efficient and obstacle-avoiding routing (EOAR) protocol that mainly focuses on the network scenarios of marine animals as obstacles. We use a fuzzy logic system to calculate the forwarding probability of each candidate forwarding relay independently by considering the propagation delay, residual energy and included angle of each relay, and we use a priority-based forwarding method to reduce packet collisions.

## 3. Models

In this section, we present the network model and underwater acoustic channel model used in the EOAR protocol.

### 3.1. Network Model

As shown in Figure 1, we consider a three-dimensional UASN where many sensor nodes coexist with marine animals. The underwater nodes communicate with each other using acoustic signals to complete the collection of oceanographic data. We define the node which generates packets as source node and the packets are transmitted through the network to the destination node. When a source node has a packet to transmit to the destination node which is far away from itself, the entire forwarding process requires many intermediate nodes to participate in forwarding packets. We define the term current node as an intermediate node which has a packet needed to forward. The current node can be an original node which just generated a new packet or a transit node which just received a packet from its neighbor node and itself is not the destination node. However, when marine animals that also use acoustic signals for daily activities are around the sensor nodes that communicate with each other, communication among sensor nodes is disturbed.

To facilitate the analysis of research objects, the entire network structure is divided into 1000 m × 1000 m × 1000 m sub-volumes. Each sub-volume contains one common sensor node. Sensor nodes $N_{j}=\left\{n_{1},n_{2},\ldots n_{j}\right\}$ are deployed in these sub-volumes according to certain deployment strategies that can increase network connectivity rate and network coverage rate. We use a node non-uniform deployment strategy [25] method to deploy the sensor nodes, in which a large number of nodes are mainly deployed in the range centered on the interference area of marine animals, and a small number of nodes are deployed in the path from the interference area to the source node and from the interference area to the destination node. Marine animals such as the Chinese white dolphins are aggregated, so there are a lot of marine animals in a certain area. To simplify the network model, we see many marine animals in a certain area as a whole (one marine animal *Ma*) to study. And the center of the area where the marine animals are located is the geographical location of the marine animal *Ma*. The marine animal *Ma* is always existed in the network and randomly moves around the network. The EOAR protocol derives the location of the marine animal *Ma* based on the probability that the marine animal will appear in a certain area. Moreover, the marine animal *Ma* does not move frequently in a certain area, and the range of movement is not large in a short time. Therefore, we can assume that the location of the marine animal *Ma* is kept in a certain area within a short period of time. It is assumed that the movement of sensor nodes caused by water currents is slow, and the sensor node is relatively stationary for a certain period of time. Sensor nodes in the network are defined as network nodes, and each sensor node can perceive, receive, and forward packets. In addition, it is also assumed that all the sensor nodes have homogenous transmission range and energy consumption. A sensor node is in the sleep state if it is not working.

### 3.2. Underwater Acoustic Channel Model

The Thorp propagation model [12] is used to describe the underwater communication, and the underwater acoustic channel gain is given as:(1)$$\;h=\frac{1}{A_{0}d^{k}\alpha {\left(f\right)}^{d}}\;$$
where *A*_0_ is a normalization constant and *d* is the distance between the current node and neighbor nodes of it. The geometry of propagation is described using the spreading factor *k*. In the proposed protocol, *k* is given as 1.5, and *f* is the signal frequency with unit kHz. The absorption coefficient *α* (f) [26], in dB/km for f in kHz, is described by the Thorp’s formula written by:(2)$$\;\alpha \left(f\right)=\{\begin{array}{c} \frac{0.11f^{2}}{1+f^{\;2}}+\frac{44f^{2}}{4100+f^{\;2}}+2.75\times 10^{-4}f^{\;2}+0.003,\;\mathrm{if}\;f\;\geq 0.4\;\\ 0.002+0.11\frac{f^{\;2}}{1+f^{\;2}}+0.011f{\;}^{2},\;\;\;\;\;\mathrm{if}\;f\;<0.4\; \end{array}\;$$

The signal-to-noise ratio (SNR) [26], which is:(3)$$\;SNR(d,\;f)\;=\;\frac{P_{s}}{A\;\left(d,\;f,\;k\right)\cdot N(f)}\;$$
$$\mathrm{and}\;A\left(d,\;f,\;k\right){\;=\;d\;}^{k}\cdot {\left(10^{\frac{\alpha \left(f\right)}{10}}\right)}^{\frac{d}{1000}},$$
where *d* represents the distance from the current node to the neighbor node and k is the spreading factor which equals to 1.5. *P_s_* is the transmit power of a sensor node. *N*(f) is the total noise which contains four kinds of noise in the acoustic channel. These four kinds of noise are turbulence *NSL_1_*(f), shipping *NSL_2_*(f), wind noise *NSL_3_*(f), and thermal noise *NSL_4_*(f), respectively. The total noise and the four kinds of noise are defined in [27,28] as follows:(4)$$\;N{\left(f\right)}_{dB}=NSL_{1}\left(f\right)+NSL_{2}\left(f\right)+NSL_{3}\left(f\right)+NSL_{4}\left(f\right),\;$$
$$\;10logNSL_{1}\left(f\right)=17-30\log\left(f\right),\;$$
$$\;10logNSL_{2}\left(f\right)=40+20\left(s-0.5\right)+26\log\left(f\right)-60\log\left(f+0.03\right),\;$$
$$\;10logNSL_{3}\left(f\right)=50+7.5\sqrt{w}+20\log\left(f\right)-40log\left(f+0.4\right),\;$$
$$\;10logNSL_{4}\left(f\right)=-15+20log\left(f\right)\;$$
where s∈(0,1) and w is the wind velocity which ranging from 0–10m/s, respectively, and f is the signal frequency (Unit: kHz).

## 4. Design of Energy-Efficient and Obstacle-Avoiding Routing Protocol

To improve packet delivery ratio and reduce energy consumption, the EOAR protocol uses geo-opportunistic routing. It assumes that sensor nodes know the geographical location of themselves and their neighbor nodes. Besides, EOAR uses three constraints which contained signal-to-noise ratio (SNR), the distance between the marine animal and the neighbor nodes of the current node, and the included angle between the current node and its neighbor nodes, to determine the candidate forwarding relay set, and then uses the fuzzy logic-based forwarding relay selection scheme to calculate the forwarding probability of the nodes in the candidate forwarding relay set. Then, EOAR uses a priority-based forwarding method to set different waiting time for the candidate forwarding relays based on their forwarding probability. The overall procedure of the proposed EOAR protocol is detailed in Figure 2.

### 4.1. Candidate Forwarding Relay Set Determination

In the EOAR protocol, when sensor node has a packet to send, the current node perceives whether there is interference from any marine animals around it and broadcasts the packet to its neighbor nodes. When the current node perceives the presence of the marine animal, the current node broadcasts the packet that contains the interference information and the time-stamp to its neighbor nodes. The interference information is the geographical location information of the marine animal. When the current node does not perceive the presence of any marine animal, the current node broadcasts the packet with only the time-stamp to its neighbor nodes. If one neighbor node receives the packet, it extracts the interference information in the packet and combines its own geographical location to calculate whether it satisfies the constraint. If one neighbor node can satisfy all the constraints, then it labels itself as a candidate forwarding relay of the current node, otherwise this neighbor node will discard the packet. The collection of all the candidate forwarding relays of the current node is called candidate forwarding relay set of the current node. Note that the candidate forwarding relay set is calculated by the neighbor nodes of the current node, not the current node itself. Since the packet of EOAR contains time-stamp information and location information, this article uses a joint time synchronization and localization design for mobile underwater sensor networks (JSL) method [29] for time synchronization and obtaining the location information.

When the current node perceives the presence of a marine animal, the first step to determine the candidate forwarding relay set is to calculate the interference area of the animal-nodes. The communication radius and the interference radius of the sensor nodes and the marine animal can obtain according to the above Equations (1) and (2). As shown in Figure 3, *R*_1_ and *R*_2_ represent the communication radius and interference radius of the marine animal, respectively. *R*_3_ and *R*_4_ represent the communication radius and interference radius of the sensor nodes, respectively.

Define the interference radius between the marine animal and sensor nodes as *R*_5_. In order to prevent interference caused by the animal, it is necessary to ensure that there is no overlap between the interference area of marine animal and the communication area of sensor nodes, and also no overlap between the interference area of the sensor node and the communication area of marine animal. Therefore, the radius of animal-nodes interference area *R*_5_ = max {*R*_2_ + *R*_3_, *R*_1_ + *R*_4_}. Consider the location of the marine animal as the center of the sphere, *R*_5_ as the radius, and then the determined spherical area is the interference area of animal-nodes. After receiving the packet with the interference information, each neighbor node of the current node determines whether it is in the interference area of animal-nodes by calculation, and the neighbor node found itself in the interference area will be in the sleep mode to further save network energy consumption.

After confirming the interference area of animal-nodes, EOAR specifies three constraints to determine the candidate forwarding relays from the neighbor nodes of the current node.
The first constrain is $SNR\geq SNR_{0}$, where $SNR_{0}$ is a preset signal to noise ratio threshold.The second constrain is that if the current node perceive interference from any marine animal, the distance $d_{s}^{j}$ from the neighbor nodes to the marine animal must satisfy the condition: $R_{5}<d_{s}^{j}\leq R_{6}$. The range covered by the area formed by the new radius (which uses *R*_5_ + 1000 as the new radius *R*_6_) may include the communication range of the neighbor nodes. The third constrain is the included angle between the current node and its neighbor nodes. For example, when the current node $n_{s}$ wants to send a packet to destination node $n_{D}$, $n_{s}$ first broadcasts the packet to its neighbor nodes $n_{s}^{j}$. Each neighbor node of the source node calculates the angle $\theta \left(n_{s}^{j}\right)$ using the cosine theorem based on the positions of the $\;n_{s}^{j}$, $n_{s}$ and $n_{D}$. The nodes $n_{s}^{j}$ with angle $\theta \left(n_{s}^{j}\right)$ < 90° meet the constraint.

The neighbor nodes of the current node that satisfy the above three constraints will be recorded in the candidate forwarding relay set. Algorithm 1 details a method of determining a candidate forwarding relay set in the situation that the current node perceives marine animal. Algorithm will be repeated in each transmission hop until the data packet finally reaches the destination node.
**Algorithm 1** Candidate Forwarding Relay Set DeterminationInput: the positions of the $n_{s}$, $n_{s}^{j}$ and $n_{D}$
Output: ***N***(*i*) = {$n_{s}^{1}$,$\;n_{s}^{2}$,…,$\;n_{s}^{j}$} //*j* is the number of the candidate forwarding relays
Initialization: *N*(*i*) = ∅;1: M = *N_n_*; // Number of neighbor nodes 2: **for**
*j* = 1: M **do**3: Calculate SNR, dsj and ∠nsjnsnD;
4: **if**
$SNR\geq SNR_{0}$, $R_{5}<d_{s}^{j}\leq R_{6}$ and $\theta \left(n_{s}^{j}\right)\;$< 90° **then**
5: ***N***(*i*) = ***N***(*i*) + {$n_{s}^{j}$};
6: **end if**
7: **end for**

If the current node cannot perceive the presence of the marine animal, then there is no interference from marine animal, thus the constraints are reduced to two, including the signal-to-noise ratio (SNR) and the included angle between the current node and its neighbor nodes. Similarly, the neighbor nodes of the current node that satisfy the above two constraints will be recorded in the candidate forwarding relay set. 

### 4.2. The Fuzzy Logic-Based Forwarding Relay Selection

In EOAR, the fuzzy logic system [30] is used to calculate the forwarding probability of the sensor nodes from the candidate forwarding relay set. Fuzzy logic system is multi-valued logic with multiple truth values, which uses continuous multiple values or membership function values between 0 and 1. Figure 4 shows the structure of fuzzy logic system which contains four parts: fuzzifier, inference engine, fuzzy rule base and defuzzifier. Commonly used fuzzy inference types are the Mamdani method and the Sugeno method. The difference between the two methods is the way to obtain the output. In this paper, Mamdani method is used and three input variables: propagation delay ratio (PDRA), cosine of the included angle (CIA) and residual energy ratio (RER) are set. Then, the fuzzy if-then rules are determined, and finally the desired output value is inferred. The centroid algorithm is used in the defuzzification phase.

The input-output mapping function *f_i_* for the fuzzy logic inference system established for each node in the candidate forwarding relay set can be expressed as:(5)$$\;P\left(n_{s}^{j}\right)=\;f_{i}\left\{Re\;\left(n_{s}^{j}\right),\;\cos\theta \left(n_{s}^{j}\right),\;P_{d}\;\left(n_{s}^{j}\right)\right\}\;$$

The specific description of the three input variables is as follows:*Re* ($n_{s}^{j}$) denotes the RER of sensor nodes. The calculation formula is as follows:(6)$$e\left(n_{s}^{j}\right)=\frac{Er\left(n_{s}^{j}\right)}{E_{I}\left(n_{s}^{j}\right)}\;$$
where $Er\left(n_{s}^{j}\right)$ and $E_{I}\left(n_{s}^{j}\right)$ are the residual energy and the initial energy of each candidate forwarding relay, respectively. $\cos\theta \left(n_{s}^{j}\right)$ represents the CIA between the current node and its neighbor nodes. $P_{d}\;\left(n_{s}^{j}\right)$ denotes the PDRA of a candidate forwarding relay. The calculation formula can be expressed as:(7)$$\;P_{d}\;\left(n_{s}^{j}\right)=\frac{Dt\left(n_{s}^{j}\right)}{D_{m}}\;$$
(8)$$\;D_{m}=\left(\frac{R_{m}}{v_{sound}}\right)\;$$
where $Dt\left(n_{s}^{j}\right)$ denotes the propagation delay from the current node to its neighbor node $n_{s}^{j}$. Specifically, the propagation delay can be obtained by subtracting the time-stamp included in the packet from the time when the data packet is received by $n_{s}^{j}$. *R_m_* is the pre-defined maximum transmission distance. The underwater acoustic speed $v_{sound}$ (m/s) is the function of depth, temperature and salinity of seawater [31]:(9)$$\begin{array}{c} \;v_{sound}=1448.96+4.591T-5.304\times 10^{-2}T^{2}+2.374\times 10^{-2}T^{3}\\ \;+1.340\left(S-35\right)+1.63\times 10^{-1}+1.675\times 10^{-7}D^{2}\\ \;-1.025\times 10^{-2}T\left(S-35\right)-7.139\times 10^{-13}TD^{3}\; \end{array}$$
where *T* is the temperature in degree Celsius, *S* is the salinity in parts per thousand, and *D* is the depth in meters. 

The input variables of fuzzy logic system and their linguistic levels are listed in Table 1.

The membership function of the triangle and the trapezoid is used in the fuzzifier to simplify the calculation. For an instance, Figure 5 shows the membership function for the RER, CIA, PDRA, respectively, which uses the triangular and the trapezoidal form. This paper defines 18 fuzzy rules used in fuzzy logic-based forwarding relay selection scheme. Table 2 shows the rules based on inference engine. According to this rule, the three variables of the input are fuzzed to obtain the fuzzy set. Finally, the fuzzy set is defuzzified to obtain a single accurate value. The accurate value of each candidate forwarding relay is expressed as the probability that the node is responsible for forwarding the packets. 

The output of the inference engine, the probability, is classified into five types: excellent, good, medium, bad, and weak. The membership functions of the probability can be seen in Figure 6. Lastly, the result of the above fuzzy operation $f_{i}\left\{Re\;\left(n_{s}^{j}\right),\;\cos\theta \left(n_{s}^{j}\right),\;P_{d}\;\left(n_{s}^{j}\right)\right\}$ is determined by using the centroid algorithm. Table 3 shows the examples of the result of the fuzzy operation. For example, if the input variables are $Re\;\left(N_{j}\right)=0.6$,$\;\cos\theta \left(N_{j}\right)=0.5$, $P_{d}\;\left(N_{j}\right)=0.4$, their corresponding membership are medium, medium and short, respectively. Then the fuzzy set is obtained according to the fuzzy logic rules, and the centroid algorithm is used to defuzzify to obtain a single accurate value, in this example, the accurate value is 0.7.

### 4.3. The Priority-Based Forwarding Method

To prevent the selected best forwarding relay from suffering failures which result in retransmissions, the current node broadcasts the data packets to all the nodes in the candidate forwarding relay set. However, two or more candidate forwarding relays may receive the same packet and repeatedly send it, resulting in a waste of energy. To reduce this waste of energy, we set a waiting time for each candidate forwarding node with the rule that nodes with larger *p* ($n_{s}^{j}$) has higher priority and lower waiting time. When a candidate forwarding relay receives a packet from the current node, it needs to overhear the forwarding status of the packet within the waiting time. If a node with a higher priority has forwarded the same data packet within the waiting time, other lower priority forwarding relays will discard this packet. 

The calculating method of the waiting time is described in detail below. Each candidate forwarding relay of the current node obtains its forwarding probability through the fuzzy logic-based forwarding relay selection method. And each of the candidate forwarding relays sets a waiting time according to its own forwarding probability. The waiting time is calculated as:(10)$$\;H_{d}\left(n_{s}^{j}\right)=\left(1-p\left(n_{s}^{j}\right)\right)\left(\frac{R_{m}}{v_{sound}}\right)+\frac{R_{m}-d\left(n_{s}^{j}\right)}{v_{sound}}\;$$
where *p* ($n_{s}^{j}$) is the probability that the sensor node is responsible for forwarding packets, *R_m_* is the pre-defined maximum transmission distance, *d* ($n_{s}^{j}$) denotes the distance between the current node and its neighbor node $n_{s}^{j}$. 

From Equation (10), it is observed that the first term reflects the relationship between the waiting time and *p* ($n_{s}^{j}$): the candidate forwarding relay with larger forwarding probability (higher priority) has less waiting time, which means that it is preferential to forward the data packet. The candidate forwarding relays overhear the packet transmission situation in their respective waiting times. If the data packet has been sent by a higher priority forwarding relay, the remaining forwarding relays discard the same data packet. The candidate forwarding relay broadcasts the data packet to the nodes in its own candidate forwarding relay set after its respective waiting time if the packet has not been discarded. Each transit hop forwarding the packets use the priority-based forwarding method until the data packet arrives the destination node.

## 5. Performance Evaluation

In this section, we evaluate the performance of the proposed EOAR protocol. The experiment verifies the packet delivery ratio, end-to-end delay, energy consumption and average network lifetime. We compare EOAR with HHVBF and VHGOR protocols.

### 5.1. Experimental Framework

In our simulation, sensor nodes are randomly deployed in a 9000 m × 9000 m × 7000 m three-dimensional underwater space. There is at most one sensor node per 1 km^3^ area. These sensor nodes are identical in every feature, such as the initial energy, communication power, transmission range and so on. Among these sensor nodes, we only set up one source–destination node pair to communicate with each other. Marine animal is also randomly active in this area. The source node generates packets follows the Poisson distribution. The transmission bit rate is 9 kb/s. The location information of marine animal accounts for 12 bytes, including latitude and longitude and depth information, the time-stamp consists of two unsigned 4-bytes integers representing seconds and nanoseconds, and the length of the packet which contains the location information of the marine animal and the time-stamp is 200 bytes. The transmission radius is 1500 m. The bottlenose dolphin is chosen as the marine animal, which occupies a frequency band of 0.52 kHz to 33 kHz [32], and the frequency used by the UASNs is 10 kHz. We use the same broadcast MAC protocol as in [16]. The experiment was simulated with Aqua-Sim [33] software, which is an underwater network simulation software based on NS-2. The simulation parameters are listed in Table 4.

### 5.2. Evaluation with Different Parameters

Packet delivery ratio (PDR) is one of the fundamental performance measures of every routing protocol. The formula for calculating PDR is given as:(11)$$\;PDR=\frac{N\left(R_{D}\right)}{N\left(G_{S}\right)},\;$$
where $N\left(R_{D}\right)$ is the number of distinctive packets received successful by the destination node, and $N\left(G_{S}\right)$ is defined as the total number of distinctive packets generated from the current node.

PDR in the network refers to the ratio between the packets received by the destination node and the packets sent by the source node. We simulate the relationship of packet delivery ratio and the number of nodes of three routing protocols.

In this simulation, the simulation duration is 500 s and the number of nodes varies between 50 and 500 with a step of 50. As shown in Figure 7, when the number of sensor nodes increases, the packet delivery ratio continues to increase. When the number of sensor nodes is only 50, PDR of the three protocols are relatively low. Since the number of sensor nodes is very small, it is impossible to form an optimal route. This may lead to packet loss during transmission, resulting in a low packet delivery ratio. Till the number of sensor nodes increases to 300, the packet delivery ratio continues to increase as the number of sensor nodes increases. This is because as the number of nodes increases, the number of neighbor nodes in the candidate forwarding relay set increases and the success ratio of transmitting the data packet from the current node to the destination node becomes higher. Therefore, adding sensor nodes will have a greater impact on network routing reachability. When the number of nodes is 300, the PDR of HHVBF and VHGOR protocols has increased by 35.9% and 17.6%, respectively. However, when the number of nodes reaches a certain value, PDR grows slower and is closer to the ideal value of 1. In this simulation, 300 can be seen as a turning point of the parameter—the number of nodes. When the number of nodes is 500, the PDR of HHVBF and VHGOR protocols has increased by 28.4% and 11.8%, respectively. Compared with HHVBF and VHGOR, EOAR always has the highest PDR with the same number of nodes. Therefore, it is observed from the evaluation that our proposed EOAR protocol can improve the packet delivery ratio compared with the other two protocols.

The end-to-end delay is expressed as the total time that a packet is sent from the source node to the destination node. The end-to-end delay primarily includes several aspects of propagation delay, transmission delay, waiting time and calculation time. The number of nodes is the same as the simulation above varies from 50 to 500 with a step of 50. The simulation duration is 500 s.

As shown in Figure 8, the end-to-end delays of the three different underwater acoustic network routing protocols are constantly changing. Compared to the HHVBF and VHGOR protocols, EOAR significantly reduces end-to-end delay. Till the number of nodes is up to 300, the end-to-end delay of the three protocols increases as the number of nodes increases. After that, the end-to-end delay of EOAR is significantly smaller than the other two protocols. The end-to-end delay of EOAR is 38.2%, 25.0% lower than that of HHVBF and VHGOR when the number of nodes is 300, respectively. This is because in spare network, there are a smaller number of potential forwarding relays. On the other hand, in a dense network, a node can find more potential forwarding relays, which takes more calculation time and waiting time than in a spare network. Moreover, under the interference of marine animals, the HHVBF protocol completes the routing that can avoid interference from marine animals is by forming a pipe for neighbor nodes of each hop and forwarding packets to all nodes in this pipe until the destination node receives the data. Therefore, the waiting time and transmission time of HHVBF are constantly increasing as the number of nodes participating in the forwarding increases. EOAR and VHGOR selects the nodes in the candidate forwarding relay set considering the interference area, which may reduce the number of nodes in the candidate forwarding relay set. EOAR and VHGOR can save the calculation time, waiting time and transmission time. Therefore, in the experiment, EOAR and VHGOR have less propagation delay than HHVBF. In addition, VHGOR does not use the method of priority-based forwarding, so when the best relay does not successfully receive the packet, retransmission occurs, causing the propagation delay to increase. This shows that the advantages of the EOAR are obvious: EOAR performs efficient routing performance when sensor nodes are densely deployed and there are marine animals in the network.

Energy consumption is also a very important indicator of network performance evaluation. The energy consumption formula in EOAR is:(12)$$\;E=\overset{M}{\underset{j=1}{\sum }}\left(E_{t_{j}}+E_{r_{j}}+E_{i_{j}}\right)\;$$
where $E_{t_{j}}$ is the energy consumption of node *N_j_* to transmit packet, and $E_{r_{j}}$ is the energy consumption of node *N_j_* to receive packet, and $E_{i_{j}}$ is the energy consumption of node *N_j_* for idle overhear mode of sensor nodes. *M* is the total number of the sensor nodes.

The energy consumption of underwater acoustic communication mainly includes the energy required to collect data, send data packets, receive data packets and the energy for calculation and idle state. In EOAR, only the nodes in candidate forwarding relay set are responsible for forwarding the packets to reduce the unnecessary node participation, thus EOAR can reduce the energy consumption. Besides, in order to prevent the same packet from being repeatedly forwarded by different nodes, EOAR sets a waiting time for each candidate forwarding relay. To further confirm that our proposed routing protocol has efficient performance, we compare its communication energy consumption with HHVBF and VHGOR. 

In this simulation, the number of sensor nodes is 300. The initial energy of each node is 10^9^ J. The power consumption for transmitting state is 10W and the power consumption for the idle state is 30 mW. It can be seen from Figure 9 that the relationship between energy consumption and time in three different routing protocols. As time increases, node energy consumption increases gradually. In the beginning, the EOAR energy consumption is higher than the VHGOR and HHVBF protocols. For example, when the simulation time is 50 s, the energy consumption of EOAR is 3.5 × 10^7^ J, and it is higher than the other two protocols. This is because the current node needs to adopt the acoustic perceive method to calculate the interference area of animal-nodes in the EOAR protocol when it encounters the interference of the marine animals. However, after a period of communication, the total energy consumption of the EOAR protocol is lower than the other two protocols because EOAR is able to avoid packet collisions and reduce the number of nodes participating in forwarding. In the simulation time of 500 s, the total energy consumption of EOAR is lower than the total energy consumption of the other two protocols. Compared with HHVBF and VHGOR protocols, the energy consumption of EOAR is reduced by 53.4% and 32.7%, respectively. This proves that the EOAR protocol has obvious advantages in reducing communication energy consumption in long time operation.

It is very difficult to replace the batteries for the sensor nodes in UASNs, so extending the network lifetime is important. In this paper, the network life cycle is defined as the duration from the network startup to the first sensor node exhausting its energy. The formula for the network lifetime is expressed as:(13)$$\;NL=\frac{\sum_{k=1}^{T}\left(EX_{k}-ST_{k}\right)}{T}\;$$
where $EX_{k}$ and $ST_{k}$ are the time at which the first sensor node exhausting its energy at the *k*th simulation run and the time at the start of the *k*th simulation run, respectively. In this simulation, the results are averaged from a total of 50 runs.

The average network lifetime was measured for the EOAR protocol and compared to the HHVBF and VHGOR protocols, as shown in Figure 10. It can be clearly seen from the figure that the network lifetime of EOAR and VHGOR are longer than that of the HHVBF protocol. This is because the HHVBF protocol only depends on the depth information of the candidate forwarding relays to select the best forwarding relay that forwards the packet, whereas the EOAR and VHGOR protocols consider the residual energy of the candidate forwarding relay in selecting the best forwarding relay. In addition, the EOAR protocol can also reduce packet collisions by setting the waiting time, so EOAR has a higher network life time than VHGOR.

## 6. Conclusions

In UASNs, avoiding obstacle interference and high energy consumption is a hot research topic for designing effective routing protocols. In this paper, we have designed an energy-efficient and obstacle-avoiding routing (EOAR) protocol. When current node wants to send a packet to the destination node, the current node first perceives the interference information of the marine animal, and then determines the interference area of the animal-nodes using the underwater acoustic channel model. The neighbor nodes of current node, which meet the three constraints label themselves as the candidate forwarding relays of the current node. According to the fuzzy logic-based forwarding relay selection scheme, which considers propagation delay, included angle and residual energy, each candidate forwarding relay calculates its forwarding probability. In addition, EOAR uses a priority-based forwarding method to prevent packet collisions in order to reduce energy consumption. Experiments show that compared with the HHVBF and VHGOR protocols, the proposed EOAR protocol in this paper has greatly improved the performance of data packet delivery ratio, end-to-end delay, energy consumption and network lifetime. In future, the channel capacity, communication reliability and other performance indicators can be integrated into the candidate forwarding relays selection of the EOAR protocol for different research targets.

## Figures and Tables

**Figure 1 sensors-18-04168-f001:**
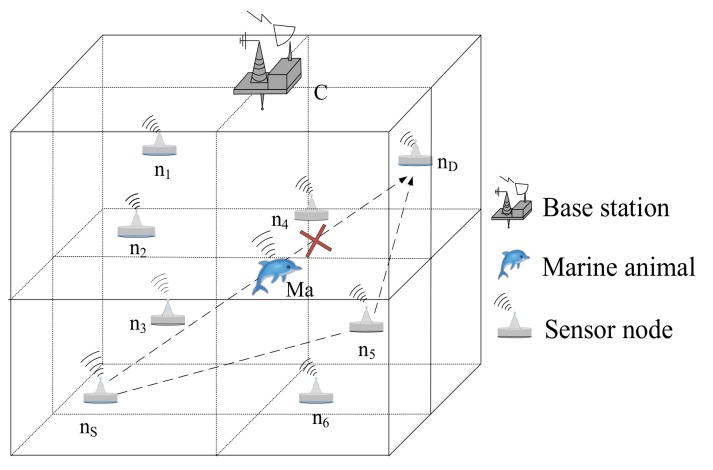
3D underwater acoustic network model.

**Figure 2 sensors-18-04168-f002:**
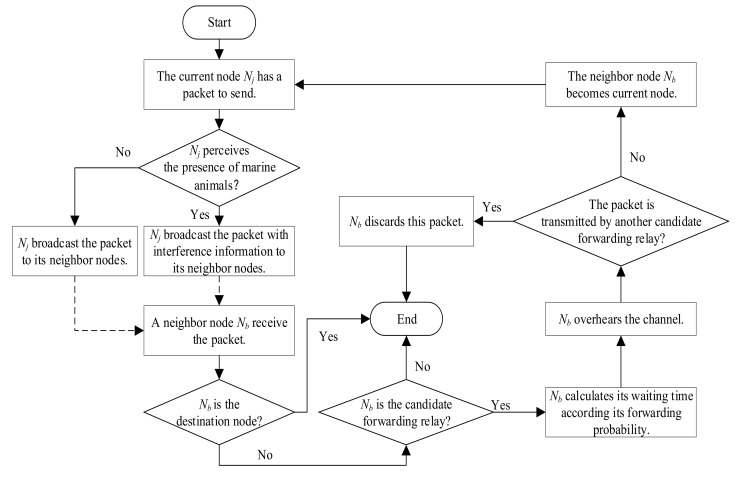
Overview of the proposed EOAR protocol.

**Figure 3 sensors-18-04168-f003:**
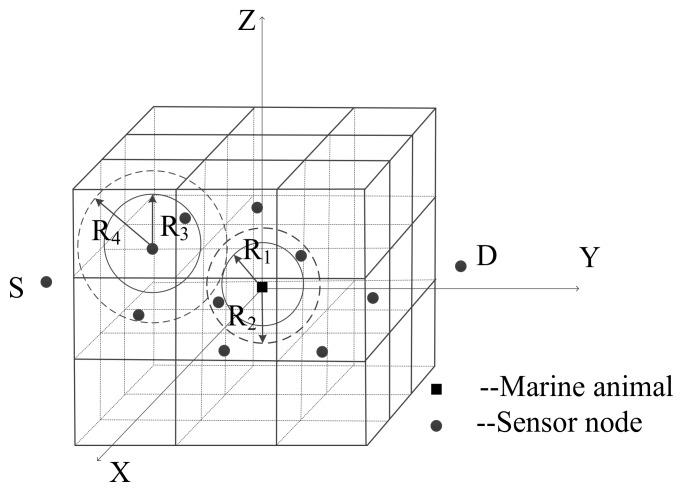
Various radius of animal and sensor.

**Figure 4 sensors-18-04168-f004:**
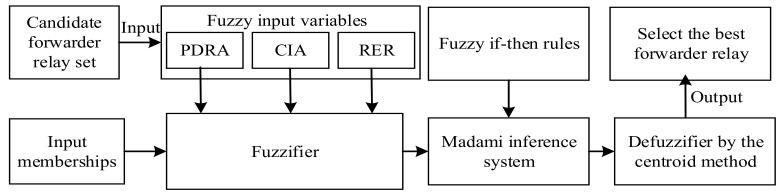
The fuzzy logic-based forwarding relay selection scheme.

**Figure 5 sensors-18-04168-f005:**
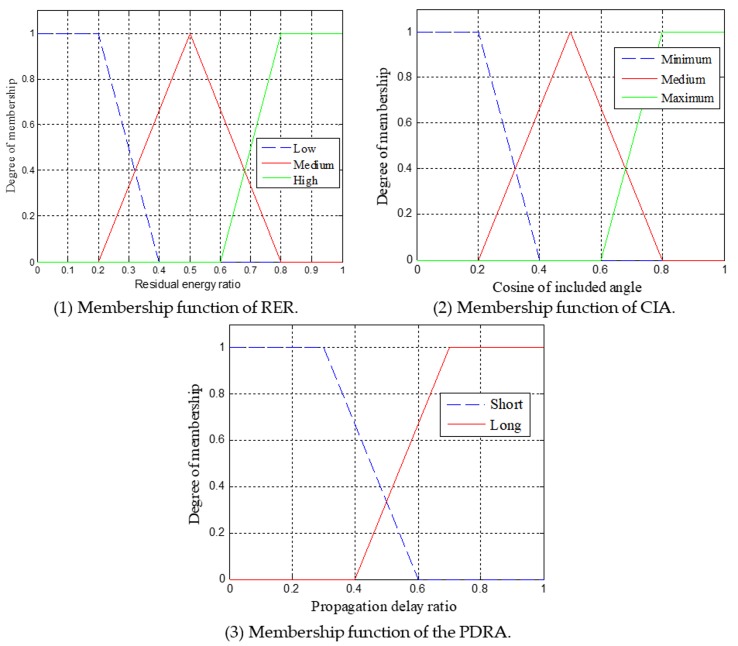
Membership functions.

**Figure 6 sensors-18-04168-f006:**
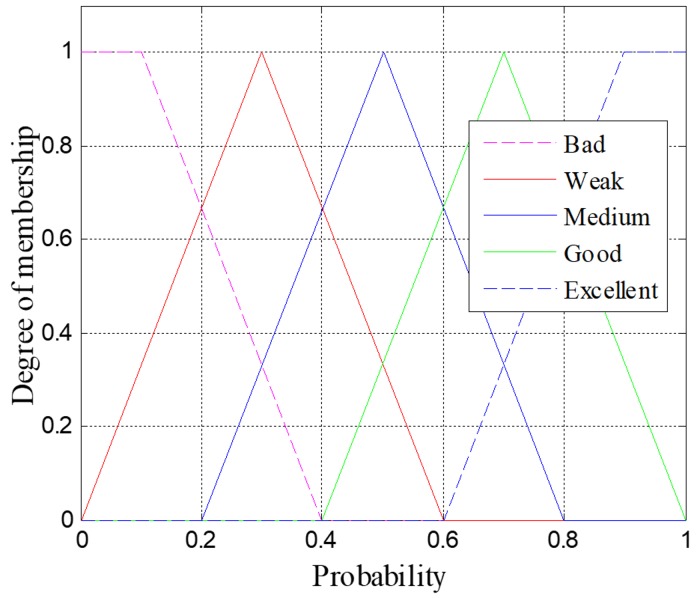
Membership function of the probability.

**Figure 7 sensors-18-04168-f007:**
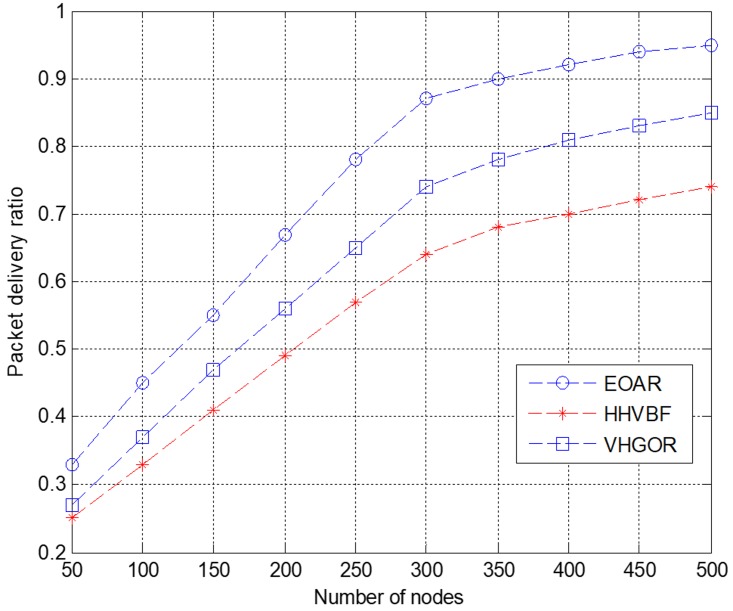
Packet delivery ratio versus the numbers of nodes.

**Figure 8 sensors-18-04168-f008:**
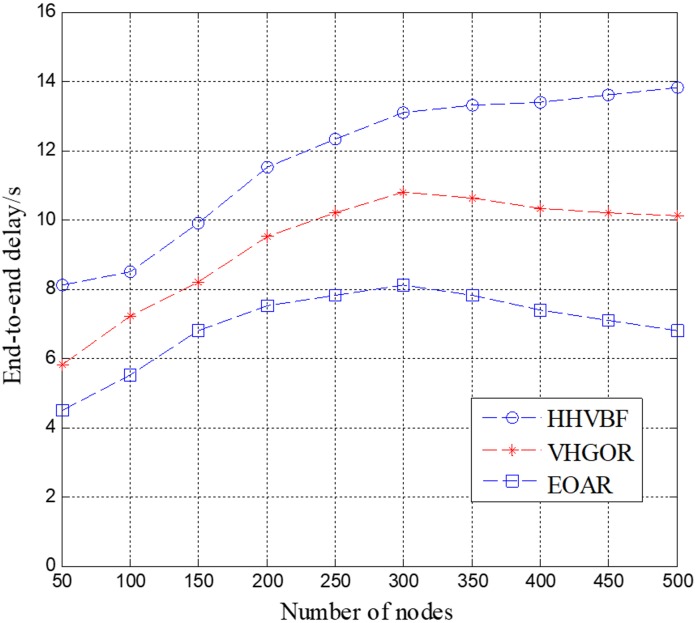
End-to-end delay versus number of nodes.

**Figure 9 sensors-18-04168-f009:**
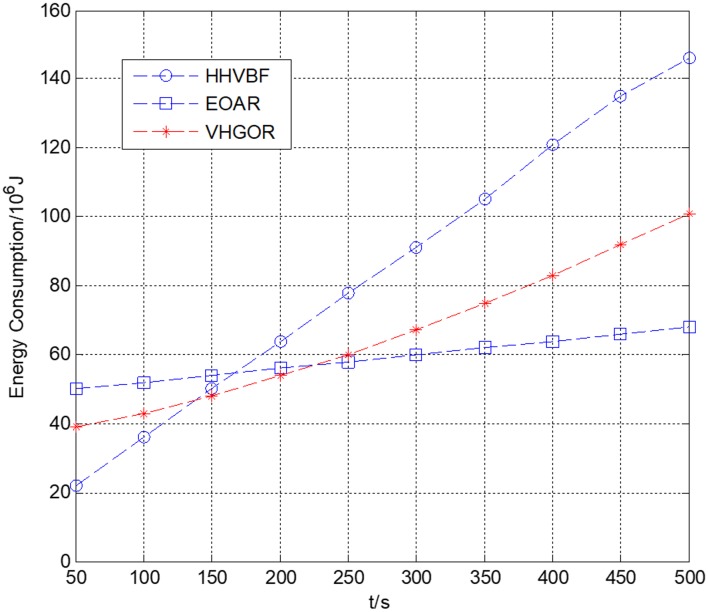
Energy consumption versus simulation time.

**Figure 10 sensors-18-04168-f010:**
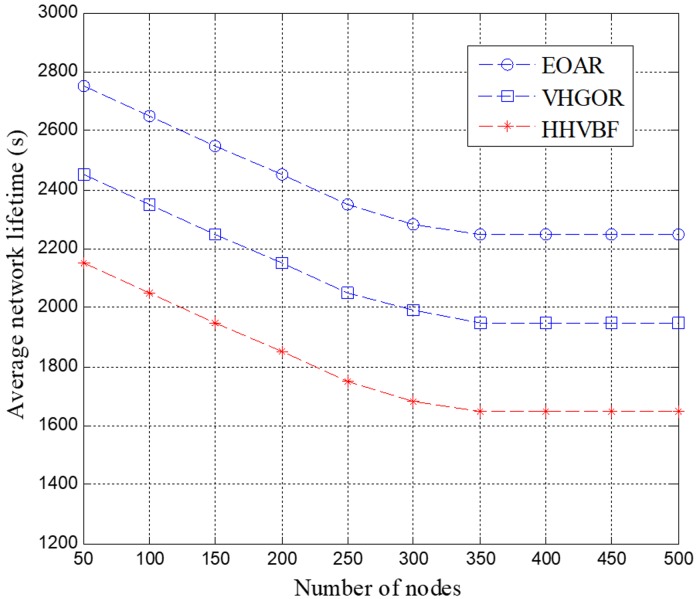
Average network lifetime versus number of nodes.

**Table 1 sensors-18-04168-t001:** Fuzzy linguistic labels of the input variables.

Input	Membership		
RER	Low	Medium	High
CIA	Minimum	Medium	Maximum
PDRA	Short	Long	−

**Table 2 sensors-18-04168-t002:** The rules based on inference engine.

(1) The linguistic level of PDRA is short.			
**RER/CIA**	**Minimum**	**Medium**	**Maximum**
Low	Medium	Medium	Weak
Medium	Good	Good	Weak
High	Excellent	Good	Weak
**(2) The linguistic level of PDRA is long.**			
**RER/CIA**	**Minimum**	**Medium**	**Maximum**
Low	Bad	Medium	Weak
Medium	Medium	Medium	Weak
High	Medium	Good	Weak

**Table 3 sensors-18-04168-t003:** The examples of the result of the fuzzy operation.

No.	$\;Re\;\left(n_{s}^{j}\right)\;$	$\;\cos\theta \left(n_{s}^{j}\right)\;$	$\;P_{d}\;\left(n_{s}^{j}\right)\;$	Probability
1	0.600	0.500	0.400	0.700
2	0.600	0.500	0.500	0.600
3	0.300	0.600	0.700	0.500
4	0.800	0.700	0.600	0.467
5	0.450	0.700	0.600	0.380

**Table 4 sensors-18-04168-t004:** Simulation parameters.

Name	Values
Simulation scene range Number of nodes Bit rate Data packet size Transmission range Transmission power Receiving power Idle power Initial energy of every sensor node	9 km × 9 km × 7 km ≤500 9 kb/s 200 B 1.5 km 10 W 3 W 30 mW 10^4^ J

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
