# Peer review of "An Energy-Efficient and Obstacle-Avoiding Routing Protocol for Underwater Acoustic Sensor Networks"

_sensors, 2018, doi:10.3390/s18124168_

Reviewer 1 Report

The proposed EOAR protocol is clearly described in the manuscript. My concerns are as follows:

The considered marine animals are not clear. In general, the frequency of marine mammals such as dolphin is less than 1 kHz, and the effect of marine animals except for marine mammals is considered as background noise. Frequencies below 1 kHz are used for long-range communication with very low data rate. In the manuscript, the frequency band is not specified. However, the data rate of 9 kbps is not feasible in frequencies below 1 kHz.

The EOAR protocol uses time-stamp, and thus the time synchronization protocol is necessary. However, the time synchronization is very difficult in underwater.

The packet overhead and routing overhead should be clearly defined. The EOAR should know interference information, time-stamp, and locations of all neighbors. Are these overheads affordable?

Fall-back mechanism is necessary.

The range of the number of nodes is not reasonable. Could the authors give examples for the employed number of nodes?

To verify the effectiveness of obstacle-avoiding routing, simulation results without marine animals should be added, and it is necessary to vary the number of marine animals.

It is assumed that the location of marine animals is known. However, it is very difficult to get the location of moving marine animals.

Author Response

Thank you very much for your comments. Please, see the attached file.

Author Response

Thanks very much for your comments. Please, see the attached file.

Round  2

Reviewer 1 Report

The comments of round 1 are throughly addressed.